# Learning Mixtures of Plackett-Luce Models from Structured Partial Orders

**Zhibing Zhao**
Department of Computer Science
Rensselaer Polytechnic Institute
Troy, NY 12180
zhaoz6@rpi.edu

**Lirong Xia**
Department of Computer Science
Rensselaer Polytechnic Institute
Troy, NY 12180
xial@cs.rpi.edu

## Abstract

Mixtures of ranking models have been widely used for heterogeneous preferences. However, learning a mixture model is highly nontrivial, especially when the dataset consists of partial orders. In such cases, the parameter of the model may not be even identifiable. In this paper, we focus on three popular structures of partial orders: ranked top-$l_1$, $l_2$-way, and choice data over a subset of alternatives. We prove that when the dataset consists of combinations of ranked top-$l_1$ and $l_2$-way (or choice data over up to $l_2$ alternatives), mixture of $k$ Plackett-Luce models is not identifiable when $l_1 + l_2 \leq 2k - 1$ ($l_2$ is set to 1 when there are no $l_2$-way orders). We also prove that under some combinations, including ranked top-3, ranked top-2 plus 2-way, and choice data over up to $4$ alternatives, mixtures of two Plackett-Luce models are identifiable. Guided by our theoretical results, we propose efficient generalized method of moments (GMM) algorithms to learn mixtures of two Plackett-Luce models, which are proven consistent. Our experiments demonstrate the efficacy of our algorithms. Moreover, we show that when full rankings are available, learning from different marginal events (partial orders) provides tradeoffs between statistical efficiency and computational efficiency.

## 1 Introduction

Suppose a group of four friends want to choose one of the four restaurants $\{a_1, a_2, a_3, a_4\}$ for dinner. The first person ranks all four restaurants as $a_2 \succ a_3 \succ a_4 \succ a_1$, where $a_2 \succ a_3$ means that "$a_2$ is strictly preferred to $a_3$". The second person says "$a_4$ and $a_3$ are my top two choices, among which I prefer $a_4$ to $a_3$". The third person ranks $a_3 \succ a_4 \succ a_1$ but has no idea about $a_2$. The fourth person has no idea about $a_4$, and would choose $a_3$ among $\{a_1, a_2, a_3\}$. How should they aggregate their preferences to choose the best restaurant?

Similar *rank aggregation* problems exist in social choice, crowdsourcing [20, 6], recommender systems [5, 3, 14, 24], information retrieval [1, 17], etc. Rank aggregation can be cast as the following statistical parameter estimation problem: given a statistical model for rank data and the agents' preferences, the parameter of the model is estimated to make decisions. Among the most widely-applied statistical models for rank aggregation are the Plackett-Luce model [19, 28] and its mixtures [8, 9, 17, 23, 30, 23]. In a Plackett-Luce model over a set of alternatives $\mathcal{A}$, each alternative is parameterized by a strictly positive number that represents its probability to be ranked higher than other alternatives. A mixture of $k$ Plackett-Luce models, denoted by $k$-PL, combines $k$ component Plackett-Luce models via the *mixing coefficients* $\vec{\alpha} = (\alpha_1, \ldots, \alpha_k) \in \mathbb{R}_{\geq 0}^k$ with $\vec{\alpha} \cdot \vec{1} = 1$, such that for any $r \leq k$, with probability $\alpha_r$, a data point is generated from the $r$-th Plackett-Luce component.

One critical limitation of Plackett-Luce model and its mixtures is that their sample space consists of *linear orders* over $\mathcal{A}$. In other words, each data point must be a full ranking of all alternatives in $\mathcal{A}$.

However, this is rarely the case in practice, because agents are often not able to rank all alternatives due to lack of information [27], as illustrated in the example in the beginning of Introduction.

In general, each rank datum is a *partial order*, which can be seen as a collection of pairwise comparisons among alternatives that satisfy transitivity. However, handling partial orders is more challenging than it appears. In particular, the pairwise comparisons of the same agent cannot be seen as independently generated due to transitivity.

Consequently, most previous works focused on *structured partial orders*, where agents' preferences share some common structures. For example, given $l \in \mathbb{N}$, in ranked-top-$l$ preferences [23, 10], agents submit a linear order over their top $l$ choices; in $l$-way preferences [21, 11, 22], agents submit a linear order over a set of $l$ alternatives, which are not necessarily their top $l$ alternatives; in choice-$l$ preferences (a.k.a. choice sets) [31], agents only specify their top choice among a set of $l$ alternatives. In particular, pairwise comparisons can be seen as 2-way preferences or choice-2 preferences.

However, as far as we know, most previous works assumed that the rank data share the same structure for their algorithms and theoretical guarantees to apply. It is unclear how rank aggregation can be done effectively and efficiently from structured partial orders of different kinds, as in the example in the beginning of Introduction. This is the key question we address in this paper.

> *How can we effectively and efficiently learn Plackett-Luce and its mixtures from structured partial orders of different kinds?*

Successfully addressing this question faces two challenges. First, to address the effectiveness concern, we need a statistical model that combines various structured partial orders to prove desirable statistical properties, and we are unaware of an existing one. Second, to address the efficiency concern, we need to design new algorithms as either previous algorithms cannot be directly applied, or it is unclear whether the theoretical guarantee such as consistency will be retained.

## 1.1 Our Contributions

Our contributions in addressing the key question are three-fold.

**Modeling Contributions.** We propose a class of statistical models to model the co-existence of the following three types of structured partial orders mentioned in the Introduction: ranked-top-$l$, $l$-way, and choice-$l$, by leveraging mixtures of Plackett-Luce models. Our models can be easily generalized to include other types of structured partial orders.

**Theoretical Contributions.** Our main theoretical results characterize the *identifiability* of the proposed models. Identifiability is fundamental in parameter estimation, which states that different parameters of the model should give different distributions over data. Clearly, if a model is non-identifiable, then no parameter estimation algorithm can be consistent.

We prove that when only ranked top-$l_1$ and $l_2$-way ($l_2$ is set to 1 if there are no $l_2$-way orders) orders are available, the mixture of $k$ Plackett-Luce models is not identifiable if $k \geq (l_1 + l_2 + 1)/2$ (Theorem 1). We also prove that the mixtures of two Plackett-Luce models is identifiable under the following combinations of structures: ranked top-3 (Theorem 2 (a) extended from [33]), ranked top-2 plus 2 way (Theorem 2 (b)), choice-2, 3, 4 (Theorem 2 (c)), and 4-way (Theorem 2 (d)). For the case of mixtures of $k$ Plackett-Luce models over $m$ alternatives, we prove that if there exist $m' \leq m$ s.t. the mixture of $k$ Plackett-Luce models over $m'$ alternatives is identifiable, we can learn the parameter using ranked top-$l_1$ and $l_2$-way orders where $l_1 + l_2 \geq m'$ (Theorem 3). This theorem, combined with Theorem 3 in [33], which provides a condition for mixtures of $k$ Plackett-Luce models to be generically identifiable, can guide the algorithm design for mixtures of arbitrary $k$ Plackett-Luce models.

**Algorithmic Contributions.** We propose efficient generalized-method-of-moments (GMM) algorithms for parameter estimation of the proposed model based on 2-PL. Our algorithm runs much faster while providing better statistical efficiency than the EM-algorithm proposed by Liu et al. [16] on datasets with large numbers of structured partial orders, see Section 6 for more details. Our algorithms are compared with the GMM algorithm by Zhao et al. [33] under two different settings. When full rankings are available, our algorithms outperform the GMM algorithm by Zhao et al. [33] in terms of MSE. When only structured partial orders are available, the GMM algorithm by Zhao et al. [33] is the best. We believe this difference is caused by the intrinsic information in the data.

## 1.2 Related Work and Discussions

**Modeling.** We are not aware of a previous model targeting rank data that consists of different types of structured partial orders. We believe that modeling the coexistence of different types of structured partial orders is highly important and practical, as it is more convenient, efficient, and accurate for an agent to report her preferences as a structured partial order of her choice. For example, some voting websites allow users to use different UIs to submit structured partial orders [4].

There are two major lines of research in rank aggregation from partial orders: learning from structured partial orders and EM algorithms for general partial orders. Popular structured partial orders investigated in the literature are pairwise comparisons [13, 12], top-$l$ [23, 10], $l$-way [21, 11, 22], and choice-$l$ [31]. Khetan and Oh [15] focused on partial orders with "separators", which is a broader class of partial orders than top-$k$. But still, [15] assumes the same structure for everyone. Our model is more general as it allows the coexistence of different types of structured partial orders in the dataset. EM algorithms have been designed for learning mixtures of Mallows' model [18] and mixtures of random utility models including the Plackett-Luce model [16], from general partial orders. Our model is less general, but as EM algorithms are often slow and it is unclear whether they are consistent, our model allows for theoretically and practically more efficient algorithms. We believe that our approach provides a principled balance between the flexibility of modeling and the efficiency of algorithms.

**Theoretical results.** Several previous works provided theoretical guarantees such as identifiability and sample complexity of mixtures of Plackett-Luce models and their extensions to structured partial orders. For linear orders, Zhao et al. [33] proved that the mixture of $k$ Plackett-Luce models over $m$ alternatives is not identifiable when $k \leq 2m - 1$ and this bound is tight for $k = 2$. We extend their results to the case of structured partial orders of various types. Ammar et al. [2, Theorem 1] proved that when $m = 2k$, where $k = 2^l$ is a nonnegative integer power of 2, there exist two different mixtures of $k$ Plackett-Luce models parameters that have the same distribution over $(2l + 1)$-way orders. Our Theorem 1 significantly extends this result in the following aspects: (i) our results includes all possible values of $k$ rather than powers of 2; (ii) we show that the model is not identifiable even under $(2^{l+1} - 1)$-way (in contrast to $(2l + 1)$-way) orders; (iii) we allow for combinations of ranked top-$l_1$ and $l_2$-way structures. Oh and Shah [26] showed that mixtures of Plackett-Luce models are in general not identifiable given partial orders, but under some conditions on the data, the parameter can be learned using pairwise comparisons. We consider many more structures than pairwise comparisons.

Recently, Chierichetti et al. [7] proved that at least $O(m^2)$ *random* marginal probabilities of partial orders are required to identify the parameter of *uniform* mixture of two Plackett-Luce models. We show that a carefully chosen set of $O(m)$ marginal probabilities can be sufficient to identify the parameter of *nonuniform* mixtures of Plackett-Luce models, which is a significant improvement. Further, our proposed algorithm can be easily modified to handle the case of uniform mixtures. Zhao et al. [35] characterized the conditions when mixtures of random utility models are generically identifiable. We focus on strict identifiability, which is stronger.

**Algorithms.** Several learning algorithms for mixtures of Plackett-Luce models have been proposed, including tensor decomposition based algorithm [26], a polynomial system solving algorithm [7], a GMM algorithm [33], and EM-based algorithms [8, 30, 23, 16]. In particular, Liu et al. [16] proposed an EM-based algorithm to learn from general partial orders. However, it is unclear whether their algorithm is consistent (as for most EM algorithms), and their algorithm is significantly slower than ours. Our algorithms for linear orders are similar to the one proposed by Zhao et al. [33], but we consider different sets of marginal probabilities and our algorithms significantly outperforms the one by Zhao et al. [33] w.r.t. MSE while taking similar running time.

## 2 Preliminaries

Let $\mathcal{A} = \{a_1, a_2, \ldots, a_m\}$ denote a set of $m$ alternatives and $\mathcal{L}(\mathcal{A})$ denote the set of all linear orders (full rankings) over $\mathcal{A}$, which are antisymmetric, transitive and total binary relations. A linear order $R \in \mathcal{L}(\mathcal{A})$ is denoted as $a_{i_1} \succ a_{i_2} \succ \ldots \succ a_{i_m}$, where $a_{i_1}$ is the most preferred alternative and $a_{i_m}$ is the least preferred alternative. A partial order $O$ is an antisymmetric and transitive binary relation. In this paper, we consider three types of strict partial orders: ranked-top-$l$ (top-$l$ for short), $l$-way, and choice-$l$, where $l \leq m$. A top-$l$ order is denoted by $O^{\text{top-}l} = [a_{i_1} \succ \ldots \succ a_{i_l} \succ \text{others}]$; an

$l$-way order is denoted by $O^{l\text{-way}} = [a_{i_1} \succ \ldots \succ a_{i_l}]$, which means that the agent does not have preferences over unranked alternatives; and a choice-$l$ order is denoted by $O^{\text{choice}-l}_{\mathcal{A}'} = (\mathcal{A}', a)$, where $\mathcal{A}' \subseteq \mathcal{A}$, $|\mathcal{A}'| = l$, and $a \in \mathcal{A}'$, which means that the agent chooses $a$ from $\mathcal{A}'$. We note that the three types of partial orders are not mutually exclusive. For example, a pairwise comparison is a 2-way order as well as a choice-2 order. Let $\mathcal{P}(\mathcal{A})$ denote the set of all partial orders of the three structures: ranked top-$l$, $l$-way, and choice-$l$ ($l \leq m$) over $\mathcal{A}$. It is worth noting that $\mathcal{L}(\mathcal{A}) \subseteq \mathcal{P}(\mathcal{A})$. Let $P = (O_1, O_2, \ldots, O_n) \in \mathcal{P}(\mathcal{A})^n$ denote the data, also called a *preference profile*. Let $O^s_{\mathcal{A}'}$ denote a partial order over a subset $\mathcal{A}'$ whose structure is $s$. When $s$ is top-$l$, $\mathcal{A}'$ is set to be $\mathcal{A}$. Let $[d]$ denote the set $\{1, 2, \ldots, d\}$.

**Definition 1.** *(Plackett-Luce model). The parameter space is $\Theta = \{\vec{\theta} = \{\theta_i | 1 \leq i \leq m, 0 < \theta_i < 1, \sum_{i=1}^m \theta_i = 1\}\}$. The sample space is $\mathcal{L}(\mathcal{A})^n$. Given a parameter $\vec{\theta} \in \Theta$, the probability of any linear order $R = [a_{i_1} \succ a_{i_2} \succ \ldots \succ a_{i_m}]$ is*

$$\Pr_{PL}(R|\vec{\theta}) = \prod_{p=1}^{m-1} \frac{\theta_{i_p}}{\sum_{q=p}^m \theta_{i_q}}.$$

Under Plackett-Luce model, a partial order $O$ can be viewed as a marginal event which consists of all linear orders that *extend $O$*, that is, for any extension $R$, $a \succ_O b$ implies $a \succ_R b$. The probabilities of the aforementioned three types of partial orders are as follows [32].

- **Top-$l$.** For any top-$l$ order $O^{\text{top-}l} = [a_{i_1} \succ \ldots \succ a_{i_l} \succ \text{others}]$, we have

$$\Pr_{\text{PL}}(O^{\text{top-}l}|\vec{\theta}) = \prod_{p=1}^{l} \frac{\theta_{i_p}}{\sum_{q=p}^m \theta_{i_q}}.$$

- **$l$-way.** For any $l$-way order $O^{l\text{-way}}_{\mathcal{A}'} = [a_{i_1} \succ \ldots \succ a_{i_l}]$, where $\mathcal{A}' = \{a_{i_1}, \ldots, a_{i_l}\}$, we have

$$\Pr_{\text{PL}}(O^{l\text{-way}}_{\mathcal{A}'}|\vec{\theta}) = \prod_{p=1}^{l-1} \frac{\theta_{i_p}}{\sum_{q=p}^{l} \theta_{i_q}}.$$

- **Choice-$l$.** For any choice order $O = (\mathcal{A}', a_i)$, we have

$$\Pr_{\text{PL}}(O|\vec{\theta}) = \frac{\theta_i}{\sum_{a_j \in \mathcal{A}'} \theta_j}.$$

In this paper, we assume that data points are i.i.d. generated from the model.

**Definition 2** (Mixtures of $k$ Plackett-Luce models for linear orders ($k$-PL)). *Given $m \geq 2$ and $k \in \mathbb{N}_+$, the sample space of $k$-PL is $\mathcal{L}(\mathcal{A})^n$. The parameter space is $\Theta = \{\vec{\theta} = (\vec{\alpha}, \vec{\theta}^{(1)}, \ldots, \vec{\theta}^{(k)})\}$, where $\vec{\alpha} = (\alpha_1, \ldots, \alpha_k)$ is the mixing coefficients. For all $r \leq k$, $\alpha_r \geq 0$ and $\sum_{r=1}^k \alpha_r = 1$. For all $1 \leq r \leq k$, $\vec{\theta}^{(r)}$ is the parameter of the $r$th Plackett-Luce component. The probability of a linear order $R$ is:*

$$\Pr_{k\text{-}PL}(R|\vec{\theta}) = \sum_{r=1}^{k} \alpha_r \Pr_{PL}(R|\vec{\theta}^{(r)}).$$

We now recall the definition of identifiability of statistical models.

**Definition 3** (Identifiability). *Let $\mathcal{M} = \{\Pr(\cdot|\vec{\theta}) : \vec{\theta} \in \Theta\}$ be a statistical model, where $\Theta$ is the parameter space and $\Pr(\cdot|\vec{\theta})$ is the distribution over the sample space associated with $\vec{\theta} \in \Theta$. $\mathcal{M}$ is identifiable if for all $\vec{\theta}, \vec{\gamma} \in \Theta$, we have*

$$\Pr(\cdot|\vec{\theta}) = \Pr(\cdot|\vec{\gamma}) \implies \vec{\theta} = \vec{\gamma}.$$

A mixture model is generally not identifiable due to the label switching problem [29], which means that labeling the components differently leads to the same distribution over data. In this paper, we consider identifiability of mixture models *modulo label switching*. That is, in Definition 3, we further require that $\vec{\theta}$ and $\vec{\gamma}$ cannot be obtained from each other by label switching.

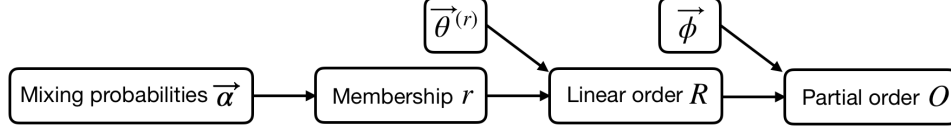

Figure 1: The mixture model for structured partial preferences.

## 3 Mixtures of Plackett-Luce Models for Partial Orders

We propose the class of mixtures of Plackett-Luce models for the aforementioned structures of partial orders. To this end, each such model should be described by the collection of allowable types of structured partial orders, denoted by $\Phi$. More precisely, $\Phi$ is a set of $u$ structures $\Phi = \{(s_1, \mathcal{A}_1), \ldots, (s_u, \mathcal{A}_u)\}$, where for any $t \in [u]$, $(s_t, \mathcal{A}_t)$ means structure $s_t$ over $\mathcal{A}_t$. For the case of top-$l$, $\mathcal{A}_t$ is set to be $\mathcal{A}$. Since the three structured considered in this paper are not mutually exclusive, **we require that $\Phi$ does not include any pair of overlapping structures simultaneously** for the model to be identifiable. There are two types of pairs of overlapping structures: (1) (top-$(m-1), \mathcal{A}$) and ($m$-way, $\mathcal{A}$); and (2) for any subset of two alternatives $\mathcal{A}'$, (2-way, $\mathcal{A}'$) and (choice-2, $\mathcal{A}'$). Each structure corresponds to a number $\phi_{\mathcal{A}_t}^{s_t} > 0$ and we require $\sum_{t=1}^{u} \phi_{\mathcal{A}_t}^{s_t} = 1$. A partial order is generated in two stages as illustrated in Figure 1: (i) a linear order $R$ is generated by $k$-PL given $\vec{\alpha}, \vec{\theta}^{(1)}, \ldots, \vec{\theta}^{(k)}$; (ii) with probability $\phi_{\mathcal{A}_t}^{s_t}$, $R$ is projected to the randomly-generated partial order structure $(s_t, \mathcal{A}_t)$ to obtain a partial order $O$. Formally, the model is defined as follows.

**Definition 4** (Mixtures of $k$ Plackett-Luce models for partial orders by $\Phi$ ($k$-PL-$\Phi$))**.** *Given $m \geq 2$, $k \in \mathbb{N}_+$, and the set of structures $\Phi = \{(s_1, \mathcal{A}_1), \ldots, (s_u, \mathcal{A}_u)\}$, the sample space is all structured partial orders defined by $\Phi$. Given $l_1 \in [m-1], l_2, l_3 \in [m]$, the parameter space is $\Theta = \{\vec{\theta} = (\vec{\phi}, \vec{\alpha}, \vec{\theta}^{(1)}, \ldots, \vec{\theta}^{(k)})\}$. The first part is a vector $\vec{\phi} = (\phi_{\mathcal{A}_1}^{s_1}, \ldots, \phi_{\mathcal{A}_u}^{s_u})$, whose entries are all positive and $\sum_{t=1}^{u} \phi_{\mathcal{A}_t}^{s_t} = 1$. The second part is $\vec{\alpha} = (\alpha_1, \ldots, \alpha_k)$ where for all $r \leq k$, $\alpha_r > 0$ and $\sum_{r=1}^{k} \alpha_r = 1$. The remaining part is $(\vec{\theta}^{(1)}, \ldots, \vec{\theta}^{(k)})$, where $\vec{\theta}^{(r)}$ is the parameter of the $r$th Plackett-Luce component. Then the probability of any partial order $O$, whose structure is defined by $(s, \mathcal{A}')$, is*

$$\Pr_{k\text{-}PL\text{-}\Phi}(O|\vec{\theta}) = \phi_{\mathcal{A}'}^{s} \sum_{r=1}^{k} \alpha_r \Pr_{PL}(O_{\mathcal{A}'}^{s}|\vec{\theta}^{(r)}).$$

For any partial order $O$ whose structure is $(s, \mathcal{A}')$, we can also write

$$\Pr_{k\text{-}PL\text{-}\Phi}(O|\vec{\theta}) = \phi_{\mathcal{A}'}^{s} \Pr_{k\text{-}PL}(O|\vec{\theta}) \tag{1}$$

where $\Pr_{k\text{-}PL}(O|\vec{\theta})$ is the marginal probability of $O$ under $k$-PL. This is a class of models because the sample space is different when $\Phi$ is different.

**Example 1.** *Let the set of alternatives be $\{a_1, a_2, a_3, a_4\}$. Consider the 2-PL-$\Phi$ $\mathcal{M}$ where $\Phi = \{(\text{top-3}, \mathcal{A}), (\text{top-2}, \mathcal{A}), (\text{3-way}, \{a_1, a_3, a_4\}), (\text{choice-3}, \{a_1, a_2, a_3\})\}$. $\phi_{\mathcal{A}}^{\text{top-3}} = 0.2$, $\phi_{\mathcal{A}}^{\text{top-2}} = 0.1$, $\phi_{\{a_1, a_3, a_4\}}^{\text{3-way}} = 0.3$, $\phi_{\{a_1, a_2, a_3\}}^{\text{choice-3}} = 0.4$, $\vec{\alpha} = [\alpha_1, \alpha_2] = [0.2, 0.8]$, $\vec{\theta}^{(1)} = [0.1, 0.2, 0.3, 0.4]$, $\vec{\theta}^{(2)} = [0.2, 0.2, 0.3, 0.3]$. Now we compute the probabilities of the following partial orders given the model: $O_1 = a_2 \succ a_3 \succ a_4 \succ a_1$ (top-3), $O_2 = a_4 \succ a_3 \succ \{a_1, a_2\}$ (top-2), $O_3 = a_3 \succ a_4 \succ a_1$ (3-way), and $O_4 = (\{a_1, a_2, a_3\}, a_3)$ (choice-3 over $\{a_1, a_2, a_3\}$). We first compute $\Pr_{PL}(O_j|\theta^{(r)})$ for all combinations of $j$ and $r$, shown in Table 1.*

| | $r = 1$ | $r = 2$ |
|---|---|---|
| $O_1$ | $\frac{0.2}{0.1+0.2+0.3+0.4} \frac{0.3}{0.1+0.3+0.4} \frac{0.4}{0.1+0.4} = 0.06$ | $\frac{0.2}{0.2+0.2+0.3+0.3} \frac{0.3}{0.2+0.3+0.3} \frac{0.3}{0.2+0.3} = 0.045$ |
| $O_2$ | $\frac{0.4}{0.1+0.2+0.3+0.4} \frac{0.3}{0.1+0.2+0.3} = 0.2$ | $\frac{0.3}{0.2+0.2+0.3+0.3} \frac{0.3}{0.2+0.2+0.3} = 0.13$ |
| $O_3$ | $\frac{0.3}{0.1+0.3+0.4} \frac{0.4}{0.1+0.4} = 0.3$ | $\frac{0.3}{0.2+0.3+0.3} \frac{0.3}{0.2+0.3} = 0.225$ |
| $O_4$ | $\frac{0.3}{0.1+0.2+0.3} = 0.5$ | $\frac{0.3}{0.2+0.2+0.3} = 0.43$ |

Table 1: $\Pr(R_j|\theta^{(r)})$ for all $j = 1, 2, 3, 4$ and $r = 1, 2$.

*Let* $\Pr_{\mathcal{M}}(O_j)$ *denote the probability of* $O_j$ *under model* $\mathcal{M}$, *we have*

$$\Pr_{\mathcal{M}}(O_1) = \phi_{\mathcal{A}}^{top\text{-}3} \sum_{r=1}^{2} \alpha_r \Pr(O_1|\vec{\theta}^{(r)}) = 0.2 \times (0.2 \times 0.06 + 0.8 \times 0.045) = 0.0096$$

$$\Pr_{\mathcal{M}}(O_2) = \phi_{\mathcal{A}}^{top\text{-}2} \sum_{r=1}^{2} \alpha_r \Pr(O_2|\vec{\theta}^{(r)}) = 0.1 \times (0.2 \times 0.2 + 0.8 \times 0.13) = 0.014$$

$$\Pr_{\mathcal{M}}(O_3) = \phi_{\{a_3,a_4\}}^{2\text{-}way} \sum_{r=1}^{2} \alpha_r \Pr(O_3|\vec{\theta}^{(r)}) = 0.3 \times (0.2 \times 0.3 + 0.8 \times 0.225) = 0.072$$

$$\Pr_{\mathcal{M}}(O_4) = \phi_{\{a_1,a_2,a_3\}}^{choice\text{-}3} \sum_{r=1}^{2} \alpha_r \Pr(O_4|\vec{\theta}^{(r)}) = 0.4 \times (0.2 \times 0.5 + 0.8 \times 0.43) = 0.18$$

## 4 (Non-)identifiability of $k$-PL-$\Phi$

Let $\Phi^{l\text{-way}} = \{(l\text{-way}, \mathcal{A}_l)|\mathcal{A}_l \in \mathcal{A}, |\mathcal{A}_l| = l\}$ and $\Phi^{\text{choice-}l} = \{(\text{choice-}l, \mathcal{A}_l)|\mathcal{A}_l \in \mathcal{A}, |\mathcal{A}_l| = l\}$. The following theorem shows that under some conditions on $\Phi$, $k$, and $m$, $k$-PL-$\Phi$ is not identifiable.

**Theorem 1.** *Given a set of* $m$ *alternatives* $\mathcal{A}$ *and any* $0 \leq l_1 \leq m-1$, $1 \leq l_2 \leq m$. *Let* $\Phi^* = \{(top\text{-}1, \mathcal{A}), \ldots, (top\text{-}l_1, \mathcal{A})\} \cup \Phi^{1\text{-way}} \cup \ldots \cup \Phi^{l_2\text{-way}}$. *Given any* $\Phi \subset \Phi^*$, *and for any* $k \geq (l_1 + l_2 + 1)/2$, $k$-PL-$\Phi$ *is not identifiable.*

We prove that the theorem holds when $\Phi = \Phi^*$. See full proof in the appendix. Considering that any $l$-way order implies a choice-$l$ order, we have the following corollary.

**Corollary 1.** *Given a set of* $m$ *alternatives* $\mathcal{A}$ *and any* $0 \leq l_1 \leq m-1$, $1 \leq l_3 \leq m$. *Let* $\Phi^* = \{(top\text{-}1, \mathcal{A}), \ldots, (top\text{-}l_1, \mathcal{A})\} \cup \Phi^{choice\text{-}1} \cup \ldots \cup \Phi^{choice\text{-}l_3}$. *Given any* $\Phi \subset \Phi^*$, *and for any* $k \geq (l_1 + l_3 + 1)/2$, $k$-PL-$\Phi$ *is not identifiable.*

Given any $k$, these results show what structures of data we cannot use if we want to interpret the learned parameter. Next, we will characterize conditions for 2-PL-$\Phi$'s to be identifiable.

**Theorem 2.** *Let* $\Phi^*$ *be one of the four combinations of structures below. For any* $\Phi \supset \Phi^*$, *2-PL-$\Phi$ over* $m \geq 4$ *alternatives is identifiable.*
*(a)* $\Phi^* = \{(top\text{-}3, \mathcal{A})\}$, *(b)* $\Phi^* = \{(top\text{-}2, \mathcal{A})\} \cup \Phi^{2\text{-way}}$, *(c)* $\Phi^* = \cup_{t=2}^{4} \Phi^{choice\text{-}t}$, *or (d)* $\Phi^* = \Phi^{4\text{-way}}$.

We first show that for any $\vec{\phi}_1 \neq \vec{\phi}_2$, the distribution over sample space must be different. Then given $\vec{\phi}$, we prove that for any $(\vec{\alpha}, \vec{\theta}^{(1)}, \ldots, \vec{\theta}^{(k)})$, there does not exist another parameter leading to the same distribution over the sample space. See the full proof in the appendix.

Identifiability for $k \geq 3$ is still an open question and Zhao et al. [33] proved that when $k \leq \lfloor \frac{m-2}{2} \rfloor!$, generic identifiability holds for $k$-PL, which means the Lebesgue measure of non-identifiable parameter is zero. We have the following theorem that can guide algorithm design for $k$-PL-$\Phi$. Full proof of Theorem 3 can be found in the appendix.

**Theorem 3.** *Let* $l_1 \in [m-1]$, $l_2 \in [m]$, *and* $\Phi^* = \{(top\text{-}l_1, \mathcal{A}), (l_2\text{-way}, \mathcal{A}')|\mathcal{A}' \in \mathcal{A}, |\mathcal{A}'| = l_2\}$. *Given any* $\Phi \supset \Phi^*$, *if* $k$-PL *over* $m'$ *alternatives is (generically) identifiable,* $k$-PL-$\Phi$ *over* $m \geq m'$ *alternatives is (generically) identifiable when* $l_1 + l_2 \geq m'$.

## 5 Consistent Algorithms for Learning 2-PL-$\Phi$

We propose a two-stage estimation algorithm. In the first stage, we make one pass of the dataset to determine $\Phi$ and estimate $\vec{\phi}$. In the second stage, we estimate the parameter $\vec{\theta}$. We note that these two stages only require one pass of the data.

In the first stage we check the existence of each structure in the dataset and estimate $\phi_{\mathcal{A}}^{top\text{-}l}$, $\phi_{\mathcal{A}'}^{l\text{-way}}$, and $\phi_{\mathcal{A}''}^{choice\text{-}l}$ for any $l$, $\mathcal{A}'$ and $\mathcal{A}''$ by dividing the occurrences of each structure by the size of the dataset. Formally, for any structure $(s, \mathcal{A}_s)$,

$$\phi_{\mathcal{A}_s}^{s} = \frac{\# \text{ of orders with structure } (s, \mathcal{A}_s)}{n} \tag{2}$$

In the second stage, we estimate $\vec{\theta}$ using the generalized-method-of-moments (GMM) algorithm. In a GMM algorithm, a set of $q$ marginal events (partial orders in the case of rank data), denoted by $E = \{\mathcal{E}_1, \ldots, \mathcal{E}_q\}$, are selected. Then $q$ *moment conditions* $\vec{g}(O, \vec{\theta}) \in \mathbb{R}^q$, which are functions of a data point $O$ and the parameter $\vec{\theta}$, are designed. The expectation of any moment condition is zero at the ground truth parameter $\vec{\theta}^*$, i.e., $E[g(O, \vec{\theta}^*)] = \vec{0}$. For a dataset $P$ with $n$ rankings, we let $\vec{g}(P, \vec{\theta}) = \frac{1}{n} \sum_{O \in P} g(O, \vec{\theta})$. Then the estimate is $\hat{\theta} = \arg\min ||g(P, \vec{\theta})||_2^2$.

Now we define moment conditions $\vec{g}(O, \vec{\theta})$. For any $t \leq q$, the $t$-th moment condition $g_t(O, \vec{\theta})$ corresponds to the event $\mathcal{E}_t$. Let $(s_t, \mathcal{A}_t)$ denote the structure of $\mathcal{E}_t$. If $O = \mathcal{E}_t$, we define $g_t(O, \vec{\theta}) = \frac{1}{\phi_{\mathcal{A}_t}^{s_t}} (\Pr_{k\text{-PL-}\Phi}(\mathcal{E}_t | \vec{\theta}) - 1)$; otherwise $g_t(O, \vec{\theta}) = \frac{1}{\phi_{\mathcal{A}_t}^{s_t}} \Pr_{k\text{-PL-}\Phi}(\mathcal{E}_t | \vec{\theta})$. Under this definition, we have

$$\vec{\theta}' = \arg\min \sum_{t=1}^{q} \left( \frac{\Pr_{k\text{-PL-}\Phi}(\mathcal{E}_t | \vec{\theta})}{\phi_{\mathcal{A}_t}^{s_t}} - \frac{\# \text{ of } \mathcal{E}_t}{n\phi_{\mathcal{A}_t}^{s_t}} \right)^2 \qquad (3)$$

We consider two ways of selecting $E$ for 2-PL-$\Phi$ guided by our Theorem 2 (b) and (c) respectively.

**Ranked top-2 and 2-way** ($\Phi = \{(\text{top-2}, \mathcal{A}), (2\text{-way}, \mathcal{A}') | \mathcal{A}' \in \mathcal{A}, |\mathcal{A}'| = 2\}$). The selected partial orders are: ranked top-2 for each pair ($m(m-1) - 1$ moment conditions) and all combinations of 2-way orders ($m(m-1)/2$ moment conditions). We remove one of the ranked top-2 orders because this corresponding moment condition is linearly dependent of the other ranked top-2 moment conditions. For the same reason, we only choose one for each 2-way comparison, resulting in $m(m-1)/2$ moment conditions. For example. in the case of $\mathcal{A} = \{a_1, a_2, a_3, a_4\}$, we can choose $E = \{a_1 \succ a_2 \succ \text{others}, a_1 \succ a_3 \succ \text{others}, a_1 \succ a_4 \succ \text{others}, a_2 \succ a_1 \succ \text{others}, a_2 \succ a_3 \succ \text{others}, a_2 \succ a_4 \succ \text{others}, a_3 \succ a_1 \succ \text{others}, a_3 \succ a_2 \succ \text{others}, a_3 \succ a_4 \succ \text{others}, a_4 \succ a_1 \succ \text{others}, a_4 \succ a_2 \succ \text{others}, a_1 \succ a_2, a_1 \succ a_3, a_1 \succ a_4, a_2 \succ a_3, a_2 \succ a_4, a_3 \succ a_4\}$.

**Choice-4.** We first group $\mathcal{A}$ into subsets of four alternatives so that $a_1$ is included in all subsets. And a small number of groups is desirable for computational considerations. One possible way is $G_1 = \{a_1, a_2, a_3, a_4\}$, $G_2 = \{a_1, a_5, a_6, a_7\}$, etc. The last group can be $\{a_1, a_{m-2}, a_{m-1}, a_m\}$. More than one overlapping alternatives across groups is fine. In this way we have $\lceil \frac{m-1}{3} \rceil$ groups. We will define $\Phi_G$ and $E_G$ for any group $G = \{a_{i_1}, a_{i_2}, a_{i_3}, a_{i_4}\}$. Then $\Phi = \cup_{t=1}^{\lceil \frac{m-1}{3} \rceil} \Phi_{G_t}$ and $E = \cup_{t=1}^{\lceil \frac{m-1}{3} \rceil} E_{G_t}$. For any $G = \{a_{i_1}, a_{i_2}, a_{i_3}, a_{i_4}\}$, $\Phi_G = \{(\text{choice-4}, G), (\text{choice-3}, G'), (\text{choice-2}, G'') | G', G'' \in G, |G'| = 3, |G''| = 2\}$. $E$ includes all 17 choice-2,3,4 orders. $E = \{(G, a_{i_1}), (G, a_{i_2}), (G, a_{i_3}), (\{a_{i_1}, a_{i_2}, a_{i_3}\}, a_{i_1}), (\{a_{i_1}, a_{i_2}, a_{i_3}\}, a_{i_2}), (\{a_{i_1}, a_{i_2}, a_{i_4}\}, a_{i_1}), (\{a_{i_1}, a_{i_2}, a_{i_4}\}, a_{i_2}), (\{a_{i_1}, a_{i_3}, a_{i_4}\}, a_{i_1}), (\{a_{i_1}, a_{i_3}, a_{i_4}\}, a_{i_3}), (\{a_{i_2}, a_{i_3}, a_{i_4}\}, a_{i_2}), (\{a_{i_2}, a_{i_3}, a_{i_4}\}, a_{i_3}), (\{a_{i_1}, a_{i_2}\}, a_{i_1}), (\{a_{i_1}, a_{i_3}\}, a_{i_1}), (\{a_{i_1}, a_{i_4}\}, a_{i_1}), (\{a_{i_2}, a_{i_3}\}, a_{i_2}), (\{a_{i_2}, a_{i_4}\}, a_{i_2}), (\{a_{i_3}, a_{i_4}\}, a_{i_3})\}$.

Formally our algorithms are collectively represented as Algorithm 1. We note that only one pass of data is required for estimating $\vec{\phi}$ and computing the frequencies of each partial order. The following theorem shows that Algorithm 1 is consistent when $E$ is chosen for "ranked top-2 and 2-way" and "choice-4".

---

**Algorithm 1** Algorithms for 2-PL-$\Phi$.

---

**Input**: Preference profile $P$ with $n$ partial orders. A set of preselected partial orders $E$.
**Output**: Estimated parameter $\vec{\theta}'$.
Estimate $\vec{\phi}$ using (2).
For each $\mathcal{E} \in E$, compute the frequency of $\mathcal{E}$.
Compute the output using (3).

---

**Theorem 4.** *Given $m \geq 4$. If there exists $\epsilon > 0$ s.t. for all $r = 1, 2$ and $i = 1, \ldots, m$, $\theta_i^{(r)} \in [\epsilon, 1]$, and $E$ is selected following either of "ranked top-2 and 2-way" and "choice-4", then Algorithm 1 is consistent.*

*Proof.* We first prove that the estimate of $\vec{\phi}$ is consistent. Let $X_t$ denote a random variable, where $X_t = 1$ if a structure $(s_t, \mathcal{A}_t)$ is observed and 0 otherwise. The dataset of $n$ partial orders is considered as $n$ trials. Let the $j$-th observation of $X_t$ be $x_j$. Then we have $E[\frac{\sum_{j=1}^{n} x_j}{n}] = \phi_{\mathcal{A}_t}^{s_t}$, which means as $n \to \infty$, $\frac{\sum_{j=1}^{n} x_j}{n}$ converges to $\phi_{\mathcal{A}_t}^{s_t}$ with probability approaching one.

Now we prove that the estimation of $\vec{\alpha}, \vec{\theta}^{(1)}, \vec{\theta}^{(2)}$ is also consistent.

We write the moment conditions $\vec{g}(P, \vec{\theta})$ as $\vec{g}_n(\vec{\theta})$ and define

$$\vec{g}_0(\vec{\theta}) = E[\vec{g}_n(\vec{\theta})].$$

Let $\vec{\theta}^*$ denote the ground truth parameter. By definition, we have

$$\vec{g}_0(\vec{\theta}^*) = E[\frac{\Pr_{k\text{-PL-}\Phi}(\mathcal{E}_t|\vec{\theta}^*)}{\phi_{\mathcal{A}_t}^{s_t}} - \frac{\#\text{ of }\mathcal{E}_t}{n\phi_{\mathcal{A}_t}^{s_t}}] = \frac{1}{\phi_{\mathcal{A}_t}^{s_t}}(\Pr_{k\text{-PL-}\Phi}(\mathcal{E}_t|\vec{\theta}^*) - E[\frac{\#\text{ of }\mathcal{E}_t}{n}]) = \vec{0}.$$

Let $Q_n(\vec{\theta}) = ||g(P, \vec{\theta})||_2^2$, which is minimized at $\vec{\theta}'$ (the estimate) and define $Q_0(\vec{\theta}) = E[Q_n(\vec{\theta})]$, which is minimized at $\vec{\theta}^*$. We first prove the following lemma:

**Lemma 1.** $\sup_{\vec{\theta}\in\Theta}|Q_n(\vec{\theta}) - Q_0(\vec{\theta})| \xrightarrow{p} 0.$

*Proof.* Recall that any moment condition $g(O_j, \vec{\theta})$ (corresponding to partial order $\mathcal{E}_t$ where $1 \leq t \leq q$) has the from $\Pr_{k\text{-PL-}\Phi}(\mathcal{E}_t|\vec{\theta}) - X_{t,j}$ where $X_{t,j} = 1$ if $\mathcal{E}_t$ is observed from $O_j$ and $X_{t,j} = 0$ otherwise. And also from $\vec{g}_n(\vec{\theta}) = \vec{g}(P, \vec{\theta}) = \frac{1}{n}\sum_{j=1}^n \vec{g}(O_j, \vec{\theta})$, for any moment condition, we have

$$|g_n(\vec{\theta}) - g_0(\vec{\theta})| = |\frac{1}{n}\sum_{j=1}^n X_{t,j} - E[X_t]| \xrightarrow{p} 0.$$

Therefore, we obtain $\sup_{\vec{\theta}\in\Theta}||\vec{g}_n(\vec{\theta}) - \vec{g}_0(\vec{\theta})|| \xrightarrow{p} \vec{0}.$

Then we have (omitting the independent variable $\vec{\theta}$)

$$|Q_n - Q_0| = |\vec{g}_n^\top \vec{g}_n - \vec{g}_0^\top \vec{g}_0| \leq |(\vec{g}_n - \vec{g}_0)^\top(\vec{g}_n - \vec{g}_0)| + 2|\vec{g}_0^\top(\vec{g}_n - \vec{g}_0)|$$

Since all moment conditions fall in $[-1, 1]$ for any $\vec{\theta} \in \Theta$, we have

$$\sup_{\vec{\theta}\in\Theta}|Q_n(\vec{\theta}) - Q_0(\vec{\theta})| \xrightarrow{p} 0.$$

$\square$

Now we are ready to prove consistency. By our Theorem 2, the model is identifiable, which means $g_0(\vec{\theta})$ is uniquely minimized at $\vec{\theta}^*$. Since $Q_0(\vec{\theta})$ is continuous and $\Theta$ is compact ($\theta_i^{(r)} \in [\epsilon, 1]$ for all $r = 0, 1$ and $i = 1, \ldots, m$), by Lemma 1 and Theorem 2.1 by Newey and McFadden [25], we have $\vec{\theta}' \xrightarrow{p} \vec{\theta}^*$. $\square$

## 6 Experiments

**Setup.** We conducted experiments on synthetic data to demonstrate the effectiveness of our algorithms. The data are generated as follows: (i) generate $\alpha, \vec{\theta}^{(1)}$, and $\vec{\theta}^{(2)}$ uniformly at random and normalize s.t. $\sum_{i=1}^m \theta_i^{(r)} = 1$ for $r = 1, 2$; (ii) generate linear orders using $k$-PL-linear; (iii) choose $\phi_{\mathcal{A}}^{\text{top-}l}$, $\phi_{\mathcal{A}'}^{l\text{-way}}$, and $\phi_{\mathcal{A}'}^{\text{choice-}l}$ and sample partial orders from the generated linear orders. The partial orders are generated from the following two models:

- ranked top-2 and 2-way: $\phi_{\mathcal{A}}^{\text{top-2}} = \frac{1}{2}$, $\phi_{\mathcal{A}'}^{\text{2-way}} = \frac{1}{m(m-1)}$ for all $\mathcal{A}' \subset \mathcal{A}$ and $|\mathcal{A}'| = 2$;

- choice-2, 3, 4: first group the alternatives as described in the previous section. Let $C = \lceil \frac{m-1}{3} \rceil$ be the number of groups. We first sample a group uniformly at random. Let $\mathcal{A}^{(4)}$ be the sampled group (of four alternatives). Then $\phi_{\mathcal{A}^{(4)}}^{\text{choice-4}} = \frac{1}{C}\frac{4}{28}$; for each subset $\mathcal{A}^{(3)} \subset \mathcal{A}^{(4)}$ of three alternatives (four such subsets within $\mathcal{A}^{(4)}$), $\phi_{\mathcal{A}^{(3)}}^{\text{choice-3}} = \frac{1}{C}\frac{3}{28}$; for each subset $\mathcal{A}^{(2)} \subset \mathcal{A}^{(4)}$ of two alternatives (six subsets within $\mathcal{A}^{(4)}$), $\phi_{\mathcal{A}^{(2)}}^{\text{choice-2}} = \frac{1}{C}\frac{1}{28}$.

Besides, we tested our algorithms on linear orders. In this case, all partial orders are marginal events of linear orders and there is no $\vec{\phi}$ estimation. Our algorithms reduce to the standard generalized-method-of-moments algorithms.

The baseline algorithms are the GMM algorithm by [33] and ELSR-Gibbs algorithm by [16]. The GMM algorithm by [33] is for linear order, but it utilizes only ranked top-3 orders. So it can be viewed as both a linear order algorithm and a partial order algorithm. We apply ELSR-Gibbs algorithm by [16] on "choice-2,3,4" datasets because the algorithm is expected to run faster than "ranked top-2 and 2-way" dataset.

All algorithms were implemented with MATLAB[1] on an Ubuntu Linux server with Intel Xeon E5 v3 CPUs each clocked at 3.50 GHz. We use Mean Squared Error (MSE), which is defined as $E[||\vec{\theta'} - \vec{\theta^*}||_2^2]$, and runtime to compare the performance of the algorithms. For fair comparisons with previous works, we ignore the $\vec{\phi}$ parameter when computing MSE.

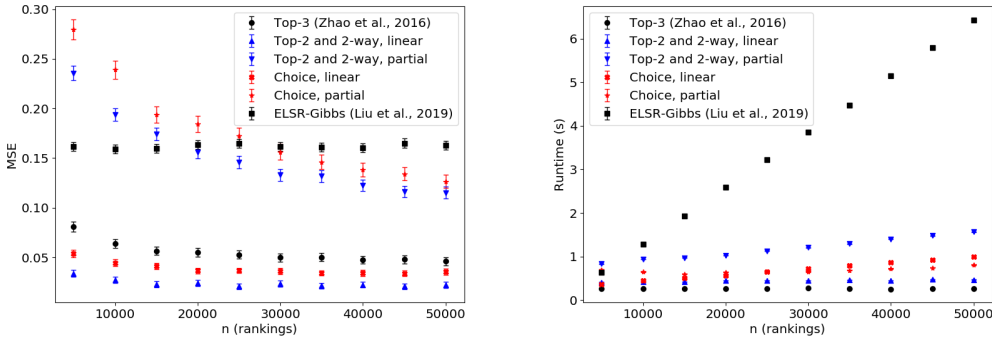

Figure 2: MSE and runtime with 95% confidence intervals for 2-PL over 10 alternatives when $n$ varies. "Choice" denotes the setting of "choice-$2, 3, 4$". For ELSR-Gibbs [16], we used the partial orders generated by "choice-$2, 3, 4$". One linear extension was generated from each partial order and three EM iterations were run. All values were averaged over 2000 trials.

**Results and Discussions.** The algorithms are compared when the number of rankings varies (Figure 2). We have the following observations.

- When learning from partial orders only: "ELSR-gibbs [16]" is much slower than other algorithms for large datasets. MSEs of all other algorithms converge towards zero as $n$ increases. We can see "top-2 and 2-way, partial" and "choice, partial" converge slower than "top-3". Ranked top-$l$ orders are generally more informative for parameter estimation than other partial orders. However, as was reported in [34], it is much more time consuming for human to pick their ranked top alternative(s) from a large set of alternatives than fully rank a small set of alternatives, which means ranked top-$l$ data are harder or more costly to collect.

- When learning from linear orders: our "ranked top-2 and 2-way, linear" and "choice-$2, 3, 4$, linear" outperform "top-3 [33]" in terms of MSE (left of Figure 2), but only slightly slower than "top-3 [33]" (Figure 2 right).

## 7 Conclusions and Future Work

We extend the mixtures of Plackett-Luce models to the class of models that sample structured partial orders and theoretically characterize the (non-)identifiability of this class of models. We propose consistent and efficient algorithms to learn mixtures of two Plackett-Luce models from linear orders or structured partial orders. For future work, we will explore more statistically and computationally efficient algorithms for mixtures of an arbitrary number of Plackett-Luce models, or the more general random utility models.

**Acknowledgments**

We thank all anonymous reviewers for helpful comments and suggestions. This work is supported by NSF #1453542 and ONR #N00014-17-1-2621.

## Footnotes

[1]Code available at `https://github.com/zhaozb08/MixPL-SPO`

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
