[Supplementary Material]

# A Proofs of Theorems in "Learning Mixtures of Plackett-Luce Models from Structured Partial Orders"

Given a set of partial orders $E$, we denote a column vector of probabilities of each partial order in $E$ for a Plackett-Luce component with parameter $\vec{\theta}^{(r)}$ by $\vec{f}_E(\vec{\theta}^{(r)})$. Given $\vec{\theta}^{(1)}, \ldots, \vec{\theta}^{(2k)}$, we define a $|E| \times 2k$ matrix $\mathbf{F}_E^k$, which is heavily used in the proofs of this paper, by $\mathbf{F}_E^k = \begin{bmatrix} \vec{f}_E(\vec{\theta}^{(1)}) & \cdots & \vec{f}_E(\vec{\theta}^{(2k)}) \end{bmatrix}$.

## A.1 Proof of Theorem 1

*Proof.* It suffices to prove that the theorem holds when $\Phi = \Phi^*$. Given $\Phi^*$, it suffices to prove that the model is not identifiable even if the $\vec{\phi}$ parameter is unique given the distribution of data.

The proof is constructive. By Lemma 1 of [33], for any $k$ and $m \geq 2k$, we only need to find $\vec{\theta}^{(1)}, \ldots, \vec{\theta}^{(2k)}$ and $\vec{\alpha} = [\alpha_1, \ldots, \alpha_{2k}]^T$ such that (1) $\mathbf{F}_E^k \cdot \vec{\alpha} = 0$, where $E$ consists of all ranked top-$l_1$ and $l_2$-way orders, and (2) $\vec{\alpha}$ has $k$ positive elements and $k$ negative elements.

We consider the case where the parameter for first alternative of $r$-th component is $e_r$, where $r = 1, 2, \ldots, k$. All other alternatives have the same parameters $b_r = \frac{1-e_r}{m-1}$.

Table 2 lists some probabilities (constant factors may be omitted). We can see the probabilities from the two classes have similar structures.

Table 2: Comparisons between two classes of moments

| $a_1$ top | $e_r$ |
|---|---|
| $a_1$ second | $\frac{e_r(1-e_r)}{e_r+m-2}$ |
| $a_1$ at position $i$ | $\frac{e_r(1-e_r)^{i-1}}{\prod_{p=1}^{i-1}(pe_r+m-1-p)}$ |
| $a_1$ not in top $l$ | $\frac{(1-e_r)^l}{\prod_{p=1}^{l-1}(pe_r+m-1-p)}$ |
| $l$-way $a_1$ top | $\frac{(m-1)e_r}{(m-l)e_r+(l-1)}$ |
| $l$-way $a_1$ second | $\frac{(m-1)e_r(1-e_r)}{((m-l)e_r+(l-1))((m-l+1)e_r+(l-2))}$ |
| $l$-way $a_1$ at position $i$ | $\frac{(m-1)e_r(1-e_r)^{i-1}}{\prod_{p=0}^{i-1}((m-l+p)e_r+(l-1-p))}$ |
| $l$-way $a_1$ at position $l$ | $\frac{(1-e_r)^{l-1}}{\prod_{p=0}^{l-2}((m-l+p)e_r+(l-1-p))}$ |

It is not hard to check that the probability for $a_1$ to be ranked at the $i$-th position in the $r$-th component is

$$\frac{(m-1)!}{(m-i)!} \frac{e_r(b_r)^{i-1}}{\prod_{p=0}^{i-1}(1-pb_r)} \tag{4}$$

where $1 \leq i \leq l_1$. The probability for $a_1$ to be ranked out of top $l_1$ position is $\frac{(m-1)!}{(m-l_1)!} \frac{(b_r)^l}{\prod_{p=0}^{l_1-1}(1-pb_r)}$.

And the probability for $a_1$ to be ranked at the $i$-th position in the $r$-th component for $l_2$-way rankings is

$$\frac{(l_2-1)!}{(l_2-i)!} \frac{e_r(b_r)^{i-1}}{\prod_{p=l_2-i}^{l_2-1}(e_r+pb_r)} \tag{5}$$

where $1 \leq i \leq l_2$.

Then $\mathbf{F}_E^k$ can be reduced to a $(l_1+l_2+1) \times (2k)$ matrix. We now define a new $(2k-1) \times (2k)$ matrix $\mathbf{H}^k$ obtained from $\mathbf{F}_E^k$ by performing the following linear operations on row vectors. (i) Make the first row of $\mathbf{H}^k$ to be $\vec{1}$; (ii) for any $2 \leq i \leq l_1+1$, the $i$-th row of $\mathbf{H}^k$ is the probability for $a_1$ to be ranked at the $(i-1)$-th position according to (4); (iii) for any $l_1+2 \leq i \leq l_1+l_2$, the $i$-th row of $\mathbf{H}^k$ is the probability for $a_1$ to be ranked at the $(i-l_1-1)$-th position in an $l_2$-way order according to (5) ; (iv) the $(l_1+l_2+1)$th row is the probability that $a_1$ is not ranked within top $l_1$; (v) remove all constant factors.

More precisely, for any $e_r$ we define the following function.

$$\vec{f}_E^*(e_r) = \begin{pmatrix} 1 \\ e_r \\ \frac{e_r(1-e_r)}{e_r+m-2} \\ \vdots \\ \frac{e_r(1-e_r)^{l_1-1}}{\prod_{p=1}^{l_1-1}(pe_r+m-1-p)} \\ \frac{(1-e_r)^{l_1}}{\prod_{p=1}^{l_1-1}(pe_r+m-1-p)} \\ \frac{e_r}{(m-l_2)e_r+(l_2-1)} \\ \vdots \\ \frac{e_r(1-e_r)^{l_2-2}}{\prod_{p=0}^{l_2-2}((m-l_2+p)e_r+(l_2-1-p))} \\ \frac{(1-e_r)^l}{\prod_{p=1}^{l-1}(pe_r+m-1-p)} \end{pmatrix}$$

Then we define $\mathbf{H}^k = [\vec{f}_E^*(e_1), \vec{f}_E^*(e_2), \cdots, \vec{f}_E^*(e_{2k})]$.

For any $r \le 2k$, let

$$\beta_r^* = \frac{\prod_{p=1}^{l_1-1}(pe_r+m-1-p) \prod_{p=0}^{l_2-2}((m-l_2+p)e_r+l_2-1-p)}{\prod_{q \ne r}(e_r-e_q)} \tag{6}$$

Note that the numerator of $\beta_r^*$ is always positive. W.l.o.g. let $e_1 < e_2 < \cdots < e_{2k}$, then half of the denominators are positive and the other half are negative. Note that the degree of the numerator of $\beta_r^*$ is $l_1 + l_2 - 2$. By Lemma 6 of [33], we have $\mathbf{H}^k \vec{\beta}^* = 0$. $\qquad\square$

## A.2  Proof of Theorem 2

*Proof.* The proof has two steps.

The first step is the same across (a), (b), (c), and (d). We show that for any $k$-PL-$\Phi$ with any parameter $\vec{\theta} = (\vec{\phi}, \vec{\alpha}, \vec{\theta}^{(1)}, \vec{\theta}^{(2)})$, there does not exist $\vec{\phi}' \ne \vec{\phi}$ s.t. for any $\vec{\theta}' = (\vec{\phi}', \vec{\alpha}', \vec{\theta}'^{(1)}, \vec{\theta}'^{(2)})$ the distribution over the sample space is exactly the same. For the purpose of contradiction suppose such $\vec{\phi}'$ exists. Since $\vec{\phi}' \ne \vec{\phi}$, there exist a structure $(s, \mathcal{A}_s)$ s.t. $\phi_{\mathcal{A}_s}^s \ne \phi_{\mathcal{A}_s}'^s$. Now we consider the total probability of all possible partial orders of this structure, denoted by $O_1, O_2, \ldots, O_w$. Then we have

$$\sum_{j=1}^w \Pr\nolimits_{k\text{-PL-}\Phi}(O_j|\vec{\theta}) = \phi_{\mathcal{A}_s}^s \ne \phi_{\mathcal{A}_s}'^s = \sum_{j=1}^w \Pr\nolimits_{k\text{-PL-}\Phi}(O_j|\vec{\theta}'),$$

which is a contradiction.

In the second step, we show that for any $k$-PL-$\Phi$ with any parameter $\vec{\theta} = (\vec{\phi}, \vec{\alpha}, \vec{\theta}^{(1)}, \vec{\theta}^{(2)})$, there does not exist $\vec{\alpha}', \vec{\theta}'^{(1)}, \vec{\theta}'^{(2)}$ s.t. for any $\vec{\theta}' = (\vec{\phi}, \vec{\alpha}', \vec{\theta}'^{(1)}, \vec{\theta}'^{(2)})$. We will prove for each of the cases (a), (b), (c), and (d).

**(a)** This step for (a) is exactly the same as the proof for [33, Theorem 2].

**(b)** We focus on $m = 4$. The case for $m > 4$ is very similar. Let $E$ consist of all ranked top-2 and 2-way orders ($\frac{3}{2}m(m-1)$ marginal probabilities). We will show that for all non-degenerate $\vec{\theta}^{(1)}, \vec{\theta}^{(2)}, \vec{\theta}^{(3)}, \vec{\theta}^{(4)}$, rank$(\mathbf{F}_E^2) = 4$. Then this part is proved by applying [33, Lemma 1].

For simplicity we use $[e_r, b_r, c_r, d_r]^\top$ to denote the parameter of $r$th Plackett-Luce model for $a_1, a_2, a_3, a_4$ respectively, i.e.,

$$\begin{bmatrix} \vec{\theta}^{(1)} & \vec{\theta}^{(2)} & \vec{\theta}^{(3)} & \vec{\theta}^{(4)} \end{bmatrix} = \begin{bmatrix} e_1 & e_2 & e_3 & e_4 \\ b_1 & b_2 & b_3 & b_4 \\ c_1 & c_2 & c_3 & c_4 \\ d_1 & d_2 & d_3 & d_4 \end{bmatrix}$$

We define $\vec{1} = [1, 1, 1, 1]$ and the following row vectors.

$$\vec{1} = [1, 1, 1, 1]$$
$$\vec{\omega}^{(1)} = [e_1, e_2, e_3, e_4]$$
$$\vec{\omega}^{(2)} = [b_1, b_2, b_3, d_3]$$
$$\vec{\omega}^{(3)} = [c_1, c_2, c_3, c_4]$$
$$\vec{\omega}^{(4)} = [d_1, d_2, d_3, d_4]$$

We have $\sum_{i=1}^{4} \vec{\omega}^{(i)} = \vec{1}$. Therefore, if there exist three $\vec{\omega}$'s such that $\{\vec{\omega}^{(1)}, \vec{\omega}^{(2)}, \vec{\omega}^{(3)}\}$ and $\vec{1}$ are linearly independent, then $\text{rank}(\mathbf{F}_E^k) = 4$. The proof is done. Because $\vec{\theta}^{(1)}, \vec{\theta}^{(2)}, \vec{\theta}^{(3)}, \vec{\theta}^{(4)}$ is non-degenerate, at least one of $\{\vec{\omega}^{(1)}, \vec{\omega}^{(2)}, \vec{\omega}^{(3)}, \vec{\omega}^{(4)}\}$ is linearly independent of $\vec{1}$. W.l.o.g. suppose $\vec{\omega}^{(1)}$ is linearly independent of $\vec{1}$. This means that not all of $e_1, e_2, e_3, e_4$ are equal. Following [33], we prove the theorem in the following two cases.

**Case 1.** $\vec{\omega}^{(2)}, \vec{\omega}^{(3)}$, and $\vec{\omega}^{(4)}$ are all linear combinations of $\vec{1}$ and $\vec{\omega}^{(1)}$.
**Case 2.** There exists a $\vec{\omega}^{(i)}$ (where $i \in \{2, 3, 4\}$) that is linearly independent of $\vec{1}$ and $\vec{\omega}^{(1)}$.

Case 2 was proved by Zhao et al. [33] using only ranked top-2 orders, as well as most of Case 1. The only remaining case is as follows. For all $r = 1, 2, 3, 4$,

$$\vec{\theta}^{(r)} = \begin{bmatrix} e_r \\ b_r \\ c_r \\ d_r \end{bmatrix} = \begin{bmatrix} e_r \\ p_2 e_r - p_2 \\ p_3 e_r - p_3 \\ -(1 + p_2 + p_3)e_r + (1 + p_2 + p_3) \end{bmatrix} \qquad (7)$$

We first show a claim, which is useful to the proof.

**Claim 1.** *Under the settings of (7), $-1 < p_2, p_3 < 0$ and there exists $p$ in $\{p_2, p_3\}$ s.t. $p \neq -\frac{1}{2}$.*

*Proof.* If $p_2 = p_3 = -\frac{1}{2}$, then $d_r = 0$, which is a contradiction. Since $e_r < 1$ and $b_r, c_r > 0$, we have $p_2, p_3 < 0$. If $p_2 \leq -1$ (or $p_3 \leq -1$), then $e_r + b_r \geq 1$ (or $e_r + c_r \geq 1$), which means parameters corresponds to all other alternatives are zero or negative. This is a contradiction. $\square$

So if $p_2 = -\frac{1}{2}$, we switch the role of $a_2$ and $a_3$. Then we have $p_2 \neq -\frac{1}{2}$.

In this case, we construct $\hat{\mathbf{F}}$ in the following way.

| $\hat{\mathbf{F}}$ | | | | Moments |
|---|---|---|---|---|
| 1 | 1 | 1 | 1 | $\vec{1}$ |
| $e_1$ | $e_2$ | $e_3$ | $e_4$ | $a_1 \succ$ others |
| $\frac{e_1 b_1}{1 - b_1}$ | $\frac{e_2 b_2}{1 - b_2}$ | $\frac{e_3 b_3}{1 - b_3}$ | $\frac{e_4 b_4}{1 - b_4}$ | $a_2 \succ a_1 \succ$ others |
| $\frac{e_1}{e_1 + b_1}$ | $\frac{e_2}{e_2 + b_2}$ | $\frac{e_3}{e_3 + b_3}$ | $\frac{e_4}{e_4 + b_4}$ | $a_1 \succ a_2$ |

Let $\vec{\omega}^{(1)} = [e_1, e_2, e_3, e_4]$.

Define $\vec{\theta}^{(b)}$

$$\vec{\theta}^{(b)} = [\frac{1}{1 - b_1}, \frac{1}{1 - b_2}, \frac{1}{1 - b_3}, \frac{1}{1 - b_4}] = [\frac{1}{1 - p_2 e_1 + p_2}, \frac{1}{1 - p_2 e_2 + p_2}, \frac{1}{1 - p_2 e_3 + p_2}, \frac{1}{1 - p_2 e_4 + p_2}]$$

And define

$$\vec{\theta}^{(be)} = [\frac{1}{(p_2 + 1)e_1 - p_2}, \frac{1}{(p_2 + 1)e_2 - p_2}, \frac{1}{(p_2 + 1)e_3 - p_2}, \frac{1}{(p_2 + 1)e_4 - p_2}]$$

Further define $\mathbf{F}^* = \begin{bmatrix} \vec{1} \\ \vec{\omega}^{(1)} \\ \vec{\theta}^{(b)} \\ \vec{\theta}^{(be)} \end{bmatrix}$. We will show $\hat{\mathbf{F}} = T^* \times \mathbf{F}^*$ where $T^*$ has full rank.

The last two rows of $\hat{\mathbf{F}}$ are

$$\frac{e_r b_r}{1 - b_r} = -e_r - \frac{1}{p_2} + \frac{1 + p_2}{p_2(1 - p_2 e_r + p_2)}$$

$$\frac{e_r}{e_r + b_r} = \frac{e_r}{(p_2 + 1)e_r - p_2} = \frac{1}{p_2 + 1} + \frac{p_2}{(p_2 + 1)((p_2 + 1)e_r - p_2)}$$

So

$$\hat{\mathbf{F}} = \begin{bmatrix} \vec{1} \\ \vec{\omega}^{(1)} \\ -\frac{1}{p_2}\vec{1} - \vec{\omega}^{(1)} + \frac{1+p_2}{p_2}\vec{\theta}^{(b)} \\ \frac{1}{p_2+1}\vec{1} + \frac{p_2}{p_2+1}\vec{\theta}^{(be)} \end{bmatrix}$$

Then we have $\hat{\mathbf{F}} = T^* \times \mathbf{F}^*$ where

$$T^* = \begin{bmatrix} 1 & 0 & 0 & 0 \\ 0 & 1 & 0 & 0 \\ -\frac{1}{p_2} & -1 & \frac{1+p_2}{p_2} & 0 \\ \frac{1}{p_2+1} & 0 & 0 & \frac{p_2}{p_2+1} \end{bmatrix}$$

From Claim 1, we have $-1 < p_2 < 0$. So $\frac{1+p_2}{p_2}, \frac{p_2}{p_2+1} \neq 0$. So $T$ has full rank. Then $\mathrm{rank}(\mathbf{F}^*) = \mathrm{rank}(\hat{\mathbf{F}})$.

If $\mathrm{rank}(\mathbf{F}_4^2) \leq 3$, then there is at least one column in $\mathbf{F}_4^2$ dependent of other columns. As all rows in $\hat{\mathbf{F}}$ are linear combinations of rows in $\mathbf{F}_4^2$, $\mathrm{rank}(\hat{\mathbf{F}}) \leq 3$. Since $\mathrm{rank}(\mathbf{F}^*) = \mathrm{rank}(\hat{\mathbf{F}})$, we have $\mathrm{rank}(\mathbf{F}^*) \leq 3$. Therefore, there exists a nonzero row vector $\vec{t} = [t_1, t_2, t_3, t_4]$, s.t.

$$\vec{t}\mathbf{F}^* = 0$$

Namely, for all $r \leq 4$,

$$t_1 + t_2 e_r + \frac{t_3}{1 - p_2 e_r + p_2} + \frac{t_4}{(p_2 + 1)e_r - p_2} = 0$$

Let

$$f(x) = t_1 + t_2 x + \frac{t_3}{1 - p_2 x + p_2} + \frac{t_4}{(p_2 + 1)e_r - p_2}$$

$$g(x) = (1 - p_2 x + p_2)((p_2 + 1)e_r - p_2)(t_1 + t_2 x) + t_3((p_2 + 1)e_r - p_2) + t_4(1 - p_2 x + p_2)$$

If any of the coefficients of $g(x)$ is nonzero, then $g(x)$ is a polynomial of degree at most 3. There will be a maximum of 3 different roots. As the equation holds for all $e_r$ where $r = 1, 2, 3, 4$. There exists $s \neq t$ s.t. $e_s = e_t$. Otherwise $g(x) = f(x) = 0$ for all $x$. We have

$$g(\frac{1 + p_2}{p_2}) = \frac{t_3(1 + 2p_2)}{p_2} = 0$$

$$g(\frac{p_2}{p_2 + 1}) = \frac{t_4 p_2}{p_2 + 1} = 0$$

From Claim 1 we know $p_2 < 0$ and $p_2 \neq -\frac{1}{2}$. So $t_3 = t_4 = 0$. Substitute it into $f(x)$ we have $f(x) = t_1 + t_2 x = 0$ for all $x$. So $t_1 = t_2 = 0$. This contradicts the nonzero requirement of $\vec{t}$. Therefore there exists $s \neq t$ s.t. $e_s = e_t$. We have $\vec{\theta}^{(s)} = \vec{\theta}^{(t)}$, which is a contradiction.

**(c)** We prove this theorem by showing that the marginal probabilities of partial orders from Theorem 2 (b) can be derived from the marginal probabilities in this theorem.

It is not hard to check the following equation holds considering any subset of four alternatives $\{a_{i_1}, a_{i_2}, a_{i_3}, a_{i_4}\}$.

$$\Pr(a_{i_1} \succ a_{i_2} \succ \{a_{i_3}, a_{i_4}\} = \Pr(a_{i_2} \succ \{a_{i_3}, a_{i_4}\}) - \Pr(a_{i_2} \succ \{a_{i_1}, a_{i_3}, a_{i_4}\})$$

The intuition is that the probability of $a_{i_2}$ being selected given $\{a_{i_2}, a_{i_3}, a_{i_4}\}$ can be decomposed into two parts: the probability of $a_{i_2}$ being selected given $\{a_{i_1}, a_{i_2}, a_{i_3}, a_{i_4}\}$ and the probability of $a_{i_2}$ being ranked at the second position and $a_{i_1}$ being ranked at the first position. This equation means we can obtain the probabilities ranked top-2 orders over a subset of four alternatives using choice data over the subset of alternatives. Then if we treat this four alternatives as a 2-PL, the parameter is identifiable.

In the case of more than four alternatives, we first group the alternatives into subsets of four and one arbitrary alternative is included in all groups. For example, when $m = 6$, we can make it two subsets: $\{a_1, a_2, a_3, a_4\}, \{a_1, a_5, a_6, a_2\}$. It is okay to have more than one overlapping alternatives, but in practice we hope to have as few groups as possible for considerations of computational efficiency. The parameter of each subset of alternatives can be uniquely learned up to a scaling factor. For any $r$, it is not hard to scale $\theta_i^{(r)}$ for all $i$ s.t. $\theta_1^{(r)}$ is the same for all groups and $\sum_{i=1}^m \theta_i^{(r)} = 1$.

**(d)** This is proved by applying the fact that any 4-way order implies a set of choice-2,3,4 orders to (c). $\qquad \square$

## A.3  Proof of Theorem 3

*Proof.* As was proved in Theorem 2, the $\vec{\phi}$ parameter is identifiable. Now we prove that $\vec{\alpha}, \vec{\theta}^{(1)}, \ldots, \vec{\theta}^{(k)}$ is (generically) identifiable.

The set of partial orders where $l_1 + l_2 = m'$ is a subset of partial orders where $l_1 + l_2 \geq m'$, so we only need to prove the cases where $l_1 + l_2 = m$. We prove this theorem by induction.

Recall that $1 \leq l_2 \leq m$. If $l_2 = 1$, then $l_1 = m - 1$, meaning this set of partial orders includes all linear rankings. The parameter is identifiable. This case serves as the base case.

Assume this theorem holds for a certain $l_1 = u$ and $l_2 = v$ where $u + v = m$, then consider the set of partial orders where $l_1 = u - 1, l_2 = v + 1$. This case adds $(v + 1)$-way orders but leaves out ranked top-$u$ orders. We can recover ranked top-$u$ rankings using ranked top-$(u - 1)$ and $(v + 1)$-way orders in the following way.

Suppose we need to recover a ranked top-$u$ order $a_1 \succ a_2 \succ \cdots \succ a_u \succ$ others. The remaining alternatives are $a_{u+1}, a_{u+1}, \cdots, a_{u+v}$. Let $U = \{a_1, a_2, \cdots, a_{u-1}\}$ and $V = \{a_{u+1}, a_{u+2}, \cdots, a_{u+v}\}$. Then we have $\Pr(a_1 \succ a_2 \succ \cdots \succ a_u \succ \text{others}) + \sum_{i=1}^{u-1} \Pr(a_u \text{ at } i\text{-th position, first } i - 1 \text{ alternatives} \in U) = \Pr(a_u \text{ to be ranked top in } \{a_u\} \cap V)$. Then the parameter can be learned in this case. $\quad\square$