[Reviews · NeurIPS 2019]

Reviewer 1



Quality Proofs of the theorems seem correct, to the best I can judge. In the experimental part, the results are given in average but no information is given about the variation of these results Clarity The paper is well organized and clearly written. The goal of the paper as well as the results are clearly presented. Significance The paper makes a significant contribution to the study of the identifiability of PL models, even if the identifiability cases are rather limited and therefore difficult to use in practice. Moreover no experience on real data is proposed.

Reviewer 2



The model seems quite novel but it is potentially too large. To handle realistic data inputs consisting of varied k-way partial orders, one needs potentially as many as min(#partial orders, m choose k) mixture components which is rather unrealistic for MLE. Perhaps investigate other models that do not require such combinatorial explosion in no. of mixture components? Theoretical results appear correct, but as mentioned above the authors should try to more fully characterize identifiability beyond the provided corner cases. It is nice to have algorithm with consistency guarantee for 2-PL. How does one extend the algorithm beyond just the data structures stated in theorem 2? Experiments and paper motivation can have more significance if tested on real partial orders. Such public datasets are readily available, so I was scratching my head as to why this isn’t used. The paper is well-organized and clear, although the notation regarding structure mixture can be slightly confusing. Related work coverage is good.

Reviewer 3



Originality, quality, and significance: Models that are able to address multiple types of ranking data simultaneously are important, and in that regard, I consider the first steps taken by this paper to be original. Some of the identifiability results presented here generalize known results for special cases, and the authors are honest in their comparisons with related work. This line of investigation is interesting, and likely to be built upon. Clarity: The writing is clear, and the proofs are correct as far as I can make out. Overall, I think that the paper makes a nice contribution to our understanding of applying PL models to multiple types of partial orders. The theoretical results are interesting, and serve as a nice first step to understanding these models. I vote to accept the paper.

[Author Response · NeurIPS 2019]

We thank all reviewers for helpful comments and feedback! Please see our responses to each reviewer's critical concerns below.

**Reviewer #2.**

- *"Standard deviation in the experimental part":* please see Figure 1. Results are shown with 95% confidence intervals. The CIs can hardly be seen in the right figure. We will update the paper with these figures.

Figure 1: MSE and runtime with $95\%$ confidence intervals.

- *"Real data study":* we feel that large-scale experiments on real-world data may not serve the purposes of this (mostly theoretical) paper, because no ground truth parameter is given in such datasets. We certainly believe that large-scale experiments on real-world datasets are important and would greatly appreciate the reviewer's suggestions on how to conduct such experiments.

**Reviewer #3.**

- *"Identifiability theorems should cover broader range of structured partial orders":* As a first step, our theorems have already significantly extended some previous work in non-trivial ways: for example, Theorem 2 covers a much broader range of partial orders than previous works ([7] and [33], as discussed in L107 and L118 of the submission). We believe that the proofs for our theorems are highly nontrivial and similar techniques do not seem to work for more general cases. Identifiability for other structures and $k \geq 3$ are certainly important (and highly challenging) future directions as the reviewer suggested.

- *"Experiments on real data and beyond mixture of 2 components"* Please see our response to Reviewer #2 on real data experiments. Identifiability of mixtures of three or more components is still an open question. Therefore, we felt that it is a little premature to run experiments on such cases. Extending the algorithm is not hard, but even the consistency of the algorithm is not known and hard to prove.

- *"Combinatorial explosion in the number of components":* Our proposed algorithms are only designed for $k = 2$. GMM algorithms for $k \geq 3$ cases need to be carefully designed w.r.t. identifiability, which is still an open question. Existing EM heuristics (as done in [16]) can be applied but their theoretical guarantee, e.g. consistency, is unclear.

- *"How does one extend the algorithm beyond just the data structures stated in Theorem 2":* We believe that new structures can be conveniently integrated into the proposed framework by introducing a parameter to represent the probability of generating the new structure. Nonetheless, it can be highly non-trivial to prove the identifiability of the new model and design consistent algorithms as we did for the most commonly-studied structures. We will add more discussions.

**Reviewer #4.** Thank you for your very encouraging comments!

[Meta-Review · NeurIPS 2019]

Congratulations! The reviewers enjoyed your paper and recommended its acceptance.